

# Quasi-particle functional Renormalisation Group calculations in the two-dimensional half-filled Hubbard model at finite temperatures

**Daniel Rohe** [1*] **and Carsten Honerkamp** [2]

**1** Forschungszentrum Juelich GmbH, Juelich Supercomputing Centre (JSC),
SimLab Quantum Materials, Juelich, Germany
**2** Institute for Theoretical Solid State Physics, RWTH Aachen University,
and JARA-FIT and CSD, 52056 Aachen, Germany

* d.rohe@fz-juelich.de

## Abstract

We present a highly parallelisable scheme for treating functional Renormalisation Group equations which incorporates a quasi-particle-based feedback on the flow and provides direct access to real-frequency self-energy data. This allows to map out the boundaries of Fermi-liquid regimes and to study the effect of quasi-particle degradation near Fermi liquid instabilities. As a first application, selected results for the two-dimensional half-filled perfectly nested Hubbard model are shown.

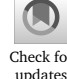

# 1   Introduction and Motivation

## 1.1   Background

Correlated electrons in solids exhibit rich behaviours at low temperatures, including technologically relevant functionalities like superconductivity, magnetism and topological states of matter. One way to approach the physics of correlated electrons theoretically are diagrammatic expansions to infinite order in the bare interactions. Since long, diagrammatic Renormalisation Group (RG) techniques had been known to sum up diagrams consistently, but their usage for correlated electrons had concentrated on zero- [1] or one-dimensional [2] situations.

    In 1996 Zanchi and Schulz triggered extensive development and usage of RG techniques in the field of correlated electrons in two dimensions with a pioneering work on the two-dimensional Hubbard model [3]. The Hubbard model (see, e.g., Ref. [4,5]) is the most prominent standard model of correlated electrons, embodying the competition between kinetic energy and repulsive interactions, and, as was recognised in the research over the decades, between various interaction-driven ordered ground states, exhibiting, e.g., magnetism, charge order and superconductivity. Understanding the Hubbard model is often considered a key to the mechanism(s) of high-temperature superconductivity in the layered cuprates [6]. In the late 1990s, after many attempts to solve the Hubbard model at stronger coupling appropriate for the high-$T_c$-cuprates, the rich physics of the model at weak to moderate interaction drew more attention, as it could be tractable by systematic diagrammatic theories and hence could give at least qualitative insights for high-temperature superconductivity [7]. While the initial study by Zanchi and Schulz [3] was motivated by the physics of high-temperature superconductors, the range of applications for functional Renormalisation Group (fRG) methods has evolved a lot in the meantime [8–12]. A variety of phenomena appearing in correlated electron systems have been treated since, with a growing set of models, geometries and technical variants of the method as such. Yet, while the fRG soon proved to be able to capture non-trivial phenomena at a qualitative level, the quantitative deviations from cases for which

other numerically reliable methods are available was and is often unsatisfactory. A number of actions have been taken to improve on this. With increasing compute power it has become possible to enhance the parametrisation of the underlying quantities such as self-energies and interactions, most recently e.g. in [13, 14], yet a complete numerical treatment which fully reflects the exact flow equation remains unfeasible. A lot of progress has been made with the explicit inclusion of frequency dependencies in one- and two-particle quantities [13–17] and in terms of a reduction in the necessary amount of spatial degrees of freedom [15, 18–20]. Furthermore, various other suggestions have been made, for example to improve on the inner consistency of the method [21] or to make explicit contact to other advanced methods such as the parquet formalism [14]. In addition, there exist numerous possible choices for regularisation and cut-off functions and also different choices of the underlying functional which can be treated, e.g. the Wick-ordered scheme, the one-particle irreducible scheme and others [8]. In short, the method today manifests itself in a zoo of methodical and technical options to choose from. This is good and bad: Good, since we may choose a variant that we can implement most easily and that suits the problem at hand. Bad, since the variants do not necessarily agree to a satisfactory degree. Qualitatively, the method has been able to provide relevant insights, but after more than 20 years the goal now is to also reach a better definition of its quantitative character.

We present a highly parallelisable, numerically simple fRG scheme that should be expandable to more complex model systems. It delivers Fermi-surface-resolved quasi-particle self-energies directly on the real-frequency axis. Quantitatively, where comparable, the data for the 2D Hubbard model below agrees quite well with other advanced state-of-the-art fRG approaches [14] that in turn tie in well with results of other numerical approaches to the Hubbard model [17]. Hence, we believe that the scheme presented here is a useful addition and can, due to its conceptual simplicity, provide useful clarifications in this field.

The same applies to recent developments of the so-called multi-loop fRG, which provides a systematic way to eliminate in particular the differences arising from different choices of cut-off functions [14].

## 1.2 Technical approach

We have decided to combine and thereby extend two fRG approaches which have long been available and may be considered somewhat brute-force, conventional or even old-fashioned in contrast to more elaborate schemes developed in the last five to ten years or so. Yet, it provides certain data which is not available (so far) within any of the more modern approaches and is complementary to these in terms of fundamental numerical approximations. As the basic scheme we chose the interaction-flow method for 1-PI correlation functions [22], and as the underlying parametric and computational description for single-particle properties, i.e. self-energies, we employ the real-frequency formalism [23]. This combination permits to incorporate, in a conceptually simple manner, the feedback of a quasi-particle weight on the fRG flow. The intention is to check in how far this technically non-trivial but formally by no means revolutionary "tweak" compares to other state-of-the art methods, be it fRG or other numerical approaches. Computing fRG-equations directly on the real axis [12, 24–26] can be considered complementary to approaches which work on the imaginary axis, be it in a continuous manner [27, 28] or as most widely used on a discrete set of Matsubara frequencies. It gives a more direct access to low-energy features which are difficult to resolve in the latter approaches due to the challenging aspects of analytical continuation. In particular, at weak-coupling some features turn out to be very subtle, as we shall see in the results section. This approach of calculating the self-energy thus has certain merits and certain draw-backs as compared to other recent evolutions and this added to our motivation to carry this aspect over into a more elaborate version of the interaction flow.

### 1.3 Outline

The main body of the article shall be concerned with the main aspects and results and is kept as clear and concise as we felt to be reasonable. Formal and technical details at the level needed to connect to existing literature are given in the appendix.

## 2 Model and Method

### 2.1 Model

We consider the one-band Hubbard model on a square lattice for Spin-$\frac{1}{2}$ Fermions in two dimensions given by

$$H = \sum_{\mathbf{j},\mathbf{j}'} \sum_{\sigma} t_{\mathbf{j}\mathbf{j}'} c^{\dagger}_{\mathbf{j}\sigma} c_{\mathbf{j}'\sigma} + U \sum_{\mathbf{j}} n_{\mathbf{j}\uparrow} n_{\mathbf{j}\downarrow}, \tag{1}$$

with a local interaction $U$ and hopping amplitudes $t_{\mathbf{j}\mathbf{j}'} = -t$ between nearest neighbors and $t_{\mathbf{j}\mathbf{j}'} = -t'$ between next-to-nearest neighbors on a square lattice. The sums run over all lattice sites $\mathbf{j}$ and spin indices $\sigma \in \uparrow, \downarrow$. The corresponding dispersion relation reads

$$\epsilon^0_{\mathbf{k}} = -2t(\cos k_x + \cos k_y) - 4t' \cos k_x \cos k_y, \tag{2}$$

and has saddle points at $\mathbf{k} = (0, \pi)$ and $(\pi, 0)$, leading to logarithmic van Hove singularities in the non-interacting density of states at energy $\epsilon_{\mathrm{vH}} = 4t'$. We measure kinetic energies with respect to the bare Fermi surface and thus use the definition

$$\xi^0_{\mathbf{k}} = -2t(\cos k_x + \cos k_y) - 4t' \cos k_x \cos k_y - \mu. \tag{3}$$

Throughout the paper we fix the energy scale by setting $t \equiv 1$. This model is a prototype-model in quantum condensed matter theory, e.g. in the context of High-temperature Superconductivity. Despite its simplicity regarding the definition it has in many respects not been solved to a satisfactory degree and remains a challenge, in particular in the region of intermediate and strong coupling strength. While in this work we set $t' = 0$ the method is not restricted to this case. Thus, we keep the definition of the model open in this respect.

### 2.2 Method

We use the interaction-flow version of the fRG, a detailed description of which as well as explicit computations of the two-particle interaction vertex for the two-dimensional Hubbard model are given in [22]. It is defined by using one-particle irreducible (1-PI) vertex functions *and* choosing a homogeneous scaling factor $g$ in the bare one-particle propagator as the flow parameter. A related scheme in the context of high-energy physics was proposed in [29]. We employ the standard truncation of the hierarchy of flow equations by setting the six-point function to zero during the flow and approximating the interaction vertex as a frequency-independent entity. In addition, we transferred the principle of calculating single-particle spectral properties directly on the real-frequency axis from the Wick-ordered scheme to the 1-PI interaction flow, allowing us to compute a quasi-particle weight from real-frequency data at all stages during the flow and to enhance the calculations by including a simple type of self-energy feedback. To this end, we compute the imaginary part of the self-energy on the real-axis with a finer grid in the low-energy region and a sufficient resolution elsewhere, which we refer to as "raw data". Subsequently, the data ist interpolated on a much finer grid via splines and the real part is calculated via Kramers-Kronig relations. A major merit of the

method is the resulting direct access to single particle spectra without any need for analytic continuation and the numerical option to refine the accuracy to a large extent. We have shifted the somewhat longish and tedious overview of the individual steps taken at the technical level to the appendix for better readability of the main article.

## 3 Results

All results shown in the following are obtained for the two-dimensional case at half filling and perfect nesting, i.e. for $t' = 0$ and $\mu = 0$, as a function of finite temperatures $T$ and the bare coupling $U$. For this case, various data from prior works are available to which we can compare, be it from other fRG schemes (for very recent samples, see [14, 17]) or other numerical methods (e.g. [30, 31]). The numerics can however directly be applied to the case $t' \neq 0$ and/or $\mu \neq 0$, which is envisaged for subsequent work. The quantities we focus on are the scale at which strong correlations set it and the corresponding single-particle self-energy on different points of the Fermi surface. The latter is computed directly on the real axis without any need for analytic continuation, which is a central virtue of the scheme. In two dimensions some unconventional features of the self-energy already arise in second order perturbation theory. These are not induced by strong correlations and we need to distinguish them from effects which are genuinely due to the evolution of correlations during the fRG flow. Therefore, we will start with simple second order perturbation theory and then move on to results obtained from the fRG treatment. The core quantities of interest will be the temperature at which the flow of the two-particle vertex diverges and the frequency dependence of the one-particle self-energy $\Sigma(\omega, \mathbf{k}_F)$, that translates into a quasi-particle weight $Z$ on the Fermi surface. The former amounts to a generalised Stoner criterion while the latter allows for direct comparison to standard Fermi-liquid theory.

### 3.1 Preliminary step: Second order bare perturbation theory

Being exact in its fundamental equation [32], functional RG turns into a perturbative method when truncations are imposed [8], as is the case in almost all its practical applications. In the form applied here it is strongly related to bare perturbation theory. Namely, the evaluation of the right-hand-side (rhs) of the flow equation at the beginning of the flow is up to a scaling factor identical to (bare) second-order perturbation theory (SOPT). Once the flow picks up its internal feedback, the fRG implicitly evolves beyond SOPT and provides information on competing instabilities and the corresponding energy scales. Thus, it is instructive and imperative to start with SOPT and use the results as a benchmark against which the fRG results can then be compared. As a reference we checked against data presented in [33], which is to the best of our knowledge the only previous work where extensive results for this case are presented.

#### 3.1.1 SOPT: Quasi-Fermi-liquid-like non-Fermi-liquid

In SOPT there is no divergence of the interaction vertex at finite temperatures, but due to the low dimensionality already in this approximation non-trivial spectral properties arise, also and in particular at finite temperatures. The raw data obtained for the imaginary part of the self-energy is shown in Figure 1, for a momentum chosen on the Fermi surface, about half way between the nodal and anti-nodal direction. A sharp down-turn appears with increasing temperature, which, by virtue of Kramers-Kronig relations, leads to an upturn, i.e. a positive slope, of the real part of the self-energy for $\omega = 0$ [33]. At $T = 0$ it is well known that in SOPT the system exhibits non-Fermi-liquid behaviour due to a linear frequency dependence (up to logarithmic corrections) of $Im\Sigma$ in the low-energy region for $\omega \to 0$ [34, 35]. At finite

temperatures however, the down-turn of $Im\Sigma$ at $\omega = 0$ implies a cross-over to a different type of non-Fermi-liquid. This is due to the upturn of $Re\Sigma$ in a narrow frequency region [36], with an expansion of very limited validity around $\omega = 0$, leading to a formal result for the $Z$-factor larger than one. In [33] a very firm statement is made that this is non-Fermi-liquid behaviour. Of course, formally this is the case in the strict sense, but the resulting spectral functions, which also generally serve as observables to be compared to experiment, very much behave like a Fermi liquid at weak coupling and low to intermediate temperatures. Hence, an alternative, pragmatic view is to say that for small values of the bare interaction $U$ the spectral function can sufficiently well be approximated over the desired range by a Fermi-liquid description which essentially neglects the narrow and limited upturn in the low-energy region. We choose to name this state "Quasi-Fermi-liquid-like non-Fermi-liquid". When $U$ becomes large enough such that the condition $\omega = U^2 Re\Sigma(\omega, U = 1.0)$ is fulfilled at finite $\omega$, the spectral function begins to deviate substantially from a Gaussian shape. Figures 2 and 3 illustrate this behaviour of $Re\Sigma$ for $T = 0.3$, the essential points being: i) For small values of $U$ the spectral function clearly has a Fermi-liquid-like shape. ii) With increasing $U$ the positive slope of $Re\Sigma$ becomes more prominent. For $U = 4$ it already leads to a dip, yet a Fermi-liquid-like Gaussian approximation remains a reasonable approximation. Even higher values, illustrated by $U = 8$, eventually lead to intersections with the line $y = \omega$ at finite $\omega$. This serves as a sensible mark for the cross-over to a two-peak structure in the spectral function which begins to render the approximation by a Fermi-liquid-like spectral function clearly questionable. iii) As long as $U$ is small enough we *can* extract a quantity which we may sensibly call "quasi-particle weight" by measuring the slope from "outside", as depicted in Figure 2. Even for $U = 4$ this is still a useful description. Of course $U = 4$ and even more so $U = 8$ are not weak coupling and the SOPT result does not mean a lot. The important message is the mechanism: In fRG couplings will grow and become large, such that the mathematical effect will be similar to what we can already see when we simply insert such number within SOPT. Here and for all fRG runs we chose a margin of $2\,T$ to either side of $\omega = 0$, which turned out reasonable. Changing this would lead to slightly different results, which is one source of variation we have to keep in mind.

Indications of such a thermal dip in SOPT, albeit in a weaker manner termed "minimum" at $\omega = 0$, were also found in [27, 37], away from half-filling and perfect nesting. While these studies did not investigate its behaviour as a function of temperature, it was observed that within fRG this effect becomes more pronounced and the difference in physical origin is emphasised, in line with our findings here, as we shall see below.

With this preparatory exercise we can now set the stage for an fRG scheme which incorporates the feedback of self-energy effects via a Fermi-liquid-like (FL-like) approximation. The catch is that we need to stay in a regime where the spectral functions are reasonably well approximated by an FL-like expression via a quasi-particle weight factor. In other words: It may happen that we leave the region of validity even before divergences in the interaction arise, and we have to take care of this. In practice, this amounts to checking the end result of an fRG flow with respect to this FL criterion. Moreover, these results from SOPT are needed to distinguish between phenomena that arise from a more elaborate fRG treatment and those that are already present in SOPT.

## 3.2 Results from Functional RG

We first provide results on the influence of the quasi-particle weight on the cross-over scale $U^*(T)$ from a weakly correlated to a strongly correlated situation, and we show the corresponding spectral functions on the Fermi surface. This scale is not sharply defined. We chose the energy scale $U^*(T)$ as the scale at which the fRG flow of the *largest absolute value* of the discretised coupling function is subject to a *maximal increase* compared to the bare interaction

by a factor of about 100.[1]

We consider the coarseness of the momentum discretisation as a parameter, albeit not in a fully comprehensive manner due to compute time limits. Thanks to a highly scalable parallel algorithm [38] we can treat flow equations for up to about 40 million couplings *and* simultaneously the two-loop contributions for the self-energy at a sufficiently accurate level.[2]

Figure 4 summarises various fRG treatments we used, together with results from other methods available in the literature [14,39–43]. We first note that omitting the particle-particle channel of the bare fRG version without any self-energy feedback at all, and keeping only the particle-hole channels, leads to lower values of $U^*(T)$, i.e. higher values of $T^*(U)$. This is expected since it amounts to a generalised (antisymmetric) RPA-like treatment. Using the full one-loop equation for the vertex, i.e. adding the particle-particle-channel, leads to a reduction of $T^*(U)$. This is instructive to notice since it helps to disentangle pure inter-channel effects from the additional influence of self-energy feedback. Adding self-energy feedback on top by means of the standard fRG via the single-scale propagator $S$ increases this suppression further. In all these cases we observed factually no dependence of the cross-over scale on the coarseness of momentum patching, within the accuracy of the definition of $U^*$ on the patching scheme. However, this changes when we switch to the so-called Katanin-corrected scheme [21], which amounts to replacing $S$ by $\frac{d}{dg}G$, i.e. the full derivative of the full propagator as a function of the flow parameter, in the flow equation of the vertex (only!). For this case we have to increase the resolution to at least 224 patches (7 shells in total in the energy direction and 32 angular sections) to obtain a sufficiently stable value for the scale of $U^*$. In relation to various transition scales found in the literature from other methods, we note that the scale as emerging from the Katanin-corrected scheme approaches these scales best. In most cases we cannot speak straight forwardly of "agreement" since the quantities looked at are all somewhat different.

The data for the critical scales match quite well those of the most recent, most advanced multi-loop fRG results via extrapolation of the antiferromagnetic susceptibility [14]. However, this is an agreement at the level of numbers and for very few points. Comparing with *one-loop* results from the same reference and similar approximations, i.e. self-energy feedback and a static vertex function, reveals differences. One would expect that the latter agrees and not the former, since the method applied here relies on the one-loop approximation as the underlying basic scheme. But such is the observation, and as stated above fRG-schemes come in many flavours, colours and shapes. Remarkably, the multi-loop scheme removes some of the quantitative variability, such that it serves as a better benchmark from a formal point of view.

The nature of the one-loop vertex flow is known to yield mean-field like divergences and is thus never associated with a true transition but rather with the onset of strong correlations, in this case antiferromagnetic ones. This motivates a common tentative view that this onset of strong correlations and the strictly valid "non"-onset of long-range order signals the transition to an unusual normal phase, e.g. the pseudogap phase. If we interpret the different scales in Figure 4 in this manner, the fRG results for the Katanin-corrected computations are much closer to the ranges found in other methods, some of which are considered numerically "exact". Yet, overall, still too little data is available to turn this observation into a robust statement, and absolute scales can be subject to various substantial uncertainties in RG calculations in general. Albeit, we can state that within the method we applied here the subsequent inclusion

---

[1]As a matter of fact the two-particle interaction vertex as a function of momenta and spin remains regular and rather small on most of its carrier, even at that stage of the flow. When speaking of a flow to strong coupling we mean that this function becomes strongly peaked and eventually develops a divergence on a sub-space of the carrier. One may even ask the question how "strong coupling" is to be properly defined. In any case the flow breaks down beyond this point.

[2]This is by no means a technical limit, but a practical one in terms of available compute resources.

of self-energy feedback as such, followed by the addition of a Katanin-correction, drives the scales up *within an otherwise unaltered fRG scheme*. We chose to not include results above $T = 0.3$, simply since already simple SOPT results in the previous section show that beyond this temperature scale the approximation we use starts to break down due to a purely thermal destruction of the "quasi"-Fermi-liquid shape of spectral functions on the Fermi surface.

Next, we check in how far the spectral functions we obtain at the end of the flow, i.e. when the couplings are already quite large, can still be described by a Fermi-liquid-like approximation and how they differ from SOPT. In Figure 5 we display the case of $T = 0.1$ and a resolution of seven angular patches between the anti-nodal and nodal direction and 15 slices in energy, amounting to 728 patches in total. Near the nodal direction we observe a stronger influence of *thermal* effects on the low-energy shape of the spectral peak, yet a weaker influence of effects due to growing interactions, since SOPT and fRG results are essentially identical. This means that we already have a peak splitting but otherwise a rather sharp peak around zero frequency. Towards the anti-nodal direction we note the opposite: The shape of the peak seems less affected by thermal effects, since both fRG and SOPT yield rather "clean" single peaks without signs of splitting. But the difference between SOPT and fRG is clearly visible in contrast to the nodal direction and shows a substantially wider peak, which can be associated with an erosion of quasi-particles. The difference and similarities of this angular dependence between fRG and SOPT becomes a little clearer when looking at $Im\Sigma$ along the Fermi surface as shown in Figure 6. The *overall* magnitude of $Im\Sigma$ in a low-energy region is higher in fRG everywhere on the Fermi surface, with a stronger effect near the anti-node than near the nodal direction. On the other hand, the *sharpness* and magnitude of the negative peak at $\omega = 0$ remains essentially unaltered. Yet, we recall that this dip becomes sharper near the nodal direction, which leads to the slight dip in the spectral function. Thus in both, fRG and SOPT, the dip is more prominent near the nodal direction, while the strong correlations entering in fRG enhance the scattering near the anti-nodal direction.

It is of central importance to pay attention to the quantity which we use in the flow to account for the effect of quasi-particle renormalisation, namely the $Z$-factor along the Fermi surface, as shown in Figure 7 . While the spectral function shows a clear change in angular dependence between SOPT and fRG, there is very little variation of the $Z$-factor along the Fermi surface even when the strongly correlated state is reached. In fact it is rather close to the SOPT results, and fRG data is even seemingly less correlated since the values are a little higher compared to the SOPT results. Thus, we infer that fRG-like correlations are active mostly via the imaginary part of the self-energy. Since the fRG is an ordinary differential equation, it is possible and instructive to also look at the derivative of the quantities of interest. At first sight it seems that any kind of pseudogap behaviour is missing in these calculations, unlike in former fRG treatments [23, 37]. This changes when we look at derived quantities *and* higher resolutions, i.e. a finer patching of momentum space. When looking deeper into the numerics there are signs of the onset of non-Fermi-liquid-like bevaviour, similar to what is observed in [23, 27]. Namely, when we work with a sufficient resolution in momentum space we can observe a change in the slope of $\frac{d}{dg}Re\Sigma$, where this slope is defined in analogy to Figure 2. We show in Figure 8 the flow of the slope of $\frac{d}{dg}Re\Sigma$ for the case of seven patches between the anti-nodal and nodal direction (i.e. 48 patches in total in the angular direction). For the patch nearest to the anti-nodal point the flow of the slope changes direction, turns upward and becomes positive when the flow enters the pseudo-critical regime. For the remaining patches the slope decreases in that area. The latter may be interpreted as the tendency to lower the quasi-particle weight, while the upturn near the anti-node signals emerging pseudogap-like features. What it also means is that some of the approximations we employ start to deviate

from some of the assumptions we make in our numerics at about $U \approx 1.6$, c.f. details in the appendix. Thus, we can only conclude that something happens in the anti-nodal direction first, which is transparent in derived quantities but not in the observable as such (derived with respect to the flow parameter). This is a generic subtle issue when interpreting fRG data: In the region of validity the physics often remains ordinary, while the region where the physics becomes interesting can lie outside the region of validity. Thus we need to look at the *transition* between the two, the access to which is also a central merit of the method as such.

In a former fluctuation-exchange approximation (FLEX/FEA) very similar structures of the spectral function were obtained, finding correlation effects and positive slopes in $Re\Sigma$, yet no pseudogap-like features [44]. Due to access to derived quantities in fRG we can additionally spot emerging tendencies. More generally, describing pseudogaps on models for interacting electrons has always been a challenge for quantum many-body methods, see e.g. [45] and therein [46, 47]. This work adds a perspective on this matter.

## 4 Conclusion and Outlook

We have combined two previous numerical implementations of fRG calculations, namely we have extended the interaction flow method [22] to include the calculation of the single-particle self-energy directly on the real-frequency axis. Based on this it was possible to include a simplified feedback of self-energy effects on the fRG-flow via the inclusion of the quasi-particle weight. The latter is a minimum requirement to apply the so-called Katanin-correction [21], which has shown to provide improvements of quantitative aspects in various cases before. The method relies on the interaction-flow cutoff which requires a patching of the whole Brillouin zone at all stages of the flow. Thus, a higher resolution is necessary and with this a higher computational need is implied, as compared to some of the other traditional cut-off methods. As an example and for an initial benchmark we obtained results for the two-dimensional Hubbard model at half filling and nearest-neighbour hopping only. The resulting temperature scales for the onset of strong correlations show improved agreement to various data from existing literature. With respect to single-particle spectra we find noticeable deviations from simple second-order perturbation theory mainly in the anti-nodal direction. Also, our calculations reveal signs of emerging pseudogap-like features in the single-particle spectral function, but only in derived quantities, i.e. for quantities on the rhs of the flow equation. Yet, this aspect is a delicate issue in fRG calculations. In some implementations pseudogap-like features appear, in others they do not. However, in those cases where these features do appear, they only show up very close to a mean-field-like critical point, such that the discrepancies are restricted to a very narrow region in the phase diagram. Parallel efforts to improve quantitative aspects of fRG calculations have also been conducted very recently in [17, 48]. The scheme presented here shows, given the approximations made, very reasonable quantitative agreement with those works regarding the pseudo-critical scales and shares the tendencies toward pseudogap opening, while on the other hand, it gives a complementary perspective from the real-frequency axis.
We feel it is important to reflect on the fact that already in second-order perturbation theory the Fermi-liquid picture breaks down with *increasing* temperature in the two-dimensional Hubbard model, an observation that has been communicated in great detail in [33] but has hardly ever been considered for comparison or as a reference, at least not to the best of our knowledge. This may limit perturbative methods *a priori* which approximate propagators at finite temperatures by free propagators.

## 4.1   Merits and Drawbacks

We summarise our view on the pros and cons of the method applied in this work. This is of course subject to discussions and maybe even taste.

### 4.1.1   Pros

- The method allows for a direct calculation of self-energies on the real-frequency axis, giving access to spectral properties without any need for analytical continuation and without the need for including a frequency dependence of the vertex.

- The scheme can extract deviations from standard behaviour at low energies, which are - for the typically relevant temperature ranges - essentially unaccessible to otherwise similar schemes that work on the Matsubara axis.

- Conceptually, the presented scheme connects transparently to simple perturbation theory and thus allows one to identify sources and non-sources of observed phenomena.

- Feedback of the quasi-particle weight on the flow of the effective interaction and the self-energy as such can be implemented, and the corresponding $Z$-factor can be calculated based on a variation of its original definition on the real-frequency axis. Other current fRG schemes work with Matsubara frequencies and require higher numerical efforts to implement such a feedback.

- The code is highly parallelised and HPC-ready, thus rather fine-grained patching schemes can be chosen, e.g. to improve the resolution of "bosonic" features in the interaction vertex. In principle, the approach can also be extended to treat more sophisticated models, but this requires additional development efforts.

- It is straight-forward to scan the phase diagram in filling and next-nearest-neighbour coupling in future works to check in how far things evolve as compared to previous calculations. In particular, the inclusion of the Katanin-correction may change things substantially due to possible pull-back effects, c.f. appendix. But this remains to be seen, the calculations are planned for the near future.

### 4.1.2   Cons

- Due to the real-frequency treatment of the self-energy there is no possibility to include any kind of frequency dependence in the interaction vertex directly.

- Some additional approximations are needed at the technical level to work with a two-loop diagram local in the flow parameter. During the flow some of these approximations become worse, which serves as an intrinsic sign to not over-interpret the results and restricts the applicability when divergences develop.

- The construction of a "quasi-Fermi-liquid" is subject to some soft choices which can imply slight variations in the results.

- It cannot be excluded that some approximations may artificially restrict the behaviour of observables during the flow. We do employ internal consistency checks, but cannot rule out such possibilities - similar to self-consistent treatments.

## 4.2 Outlook

In terms of immediate applications we envisage the more general case of finite doping and finite next-nearest neighbour hopping $t'$ in order to study transition scales and spectral properties near other types of order, such as superconductivity, ferromagnetism, etc. It is also straight forward to extract data for the self-energy on the imaginary axis in order to compare to other methods, be it fRG variants, DCA-based schemes, cDMFT calculations, etc. As mentioned before, we feel that an incremental view with respect to the level of approximation needs to be emphasised. We saw that already second-order perturbation theory yields aspects which can be quite close to the result of more elaborate schemes. It is thus helpful to always consider it as a benchmark to better understand and interpret the origins (and non-origins) of physical observations.

Another goal will be to extend this scheme to few-band systems, in particular to models of unconventional superconductors like quasi-two-dimensional iron arsenides or chalcogenides, or strontium ruthenates. For such systems, Fermi-surface-resolved quasiparticle properties are of great interest [49,50] and are, in current theory, investigated quite phenomenologically [51] or by techniques complementary to ours [50].

## Figures

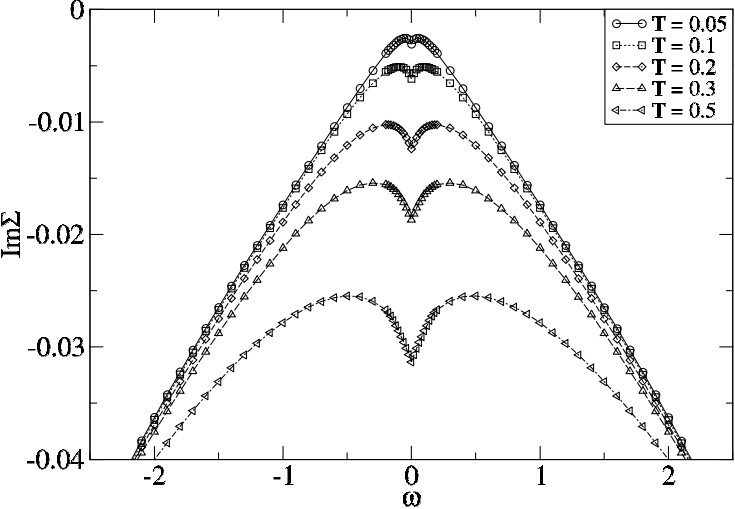

Figure 1: Raw data for $Im\Sigma(\omega, \mathbf{k}_F)$ at $U = 1.0$ in second order bare perturbation theory. $\mathbf{k}_F$ is chosen about half way between the nodal point $(\pi/2, \pi/2)$ and the anti-nodal point $(\pi, 0)$. The down-turn at $\omega = 0$ increases with temperature and leads to a positive slope of $Re\Sigma$ at $\omega = 0$.

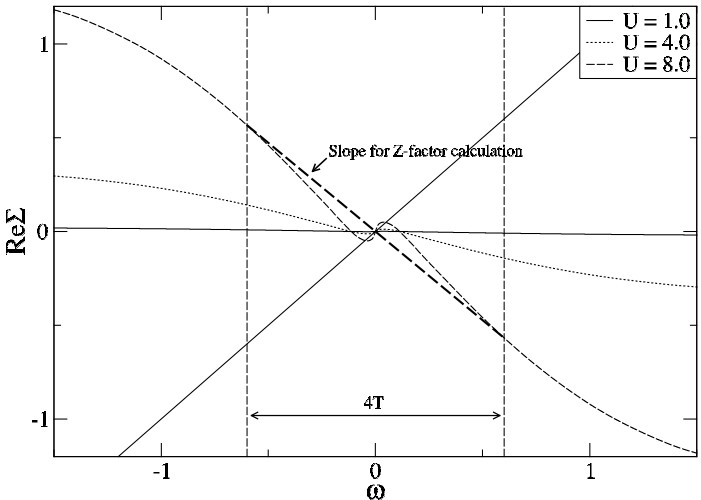

Figure 2: $Re\Sigma$ in SOPT at $T = 0.3$ for increasing values of $U$, corresponding to Figure 1. For $U = 8$ the spectral function aquires a double-peak structure due to the intersection of $Re\Sigma$ with $y = \omega$, as shown in Figure 3. For the purpose of calculating the $Z$-factor in the fRG flow to mimic the quasi-Fermi-liquid-like behaviour, the slope is numerically determined from well outside of the region of inverted slope.

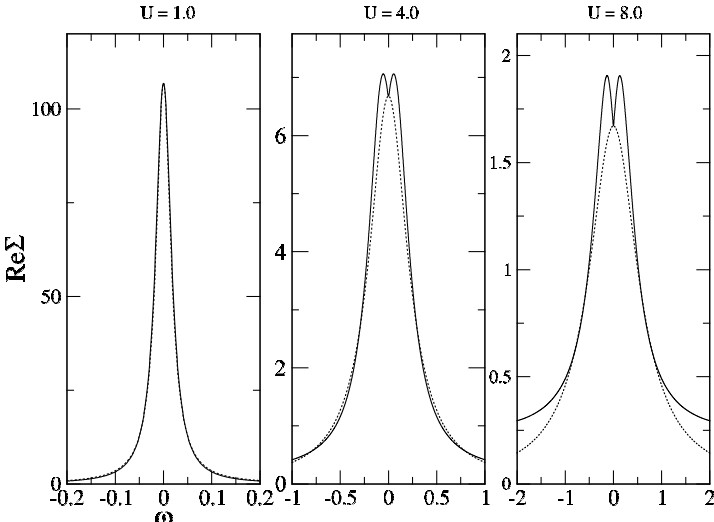

Figure 3: Spectral functions from second-order perturbation theory (SOPT) corresponding to Figure 2. The lightly dotted line shows the Fermi-liquid (i.e. Gaussian) approximation for the spectral function using a quasi-particle weight which ignores the upturn of $Re(\Sigma)$. It is constrained to have the same value as the full spectral function at $\omega = 0$. Mind the different scales!

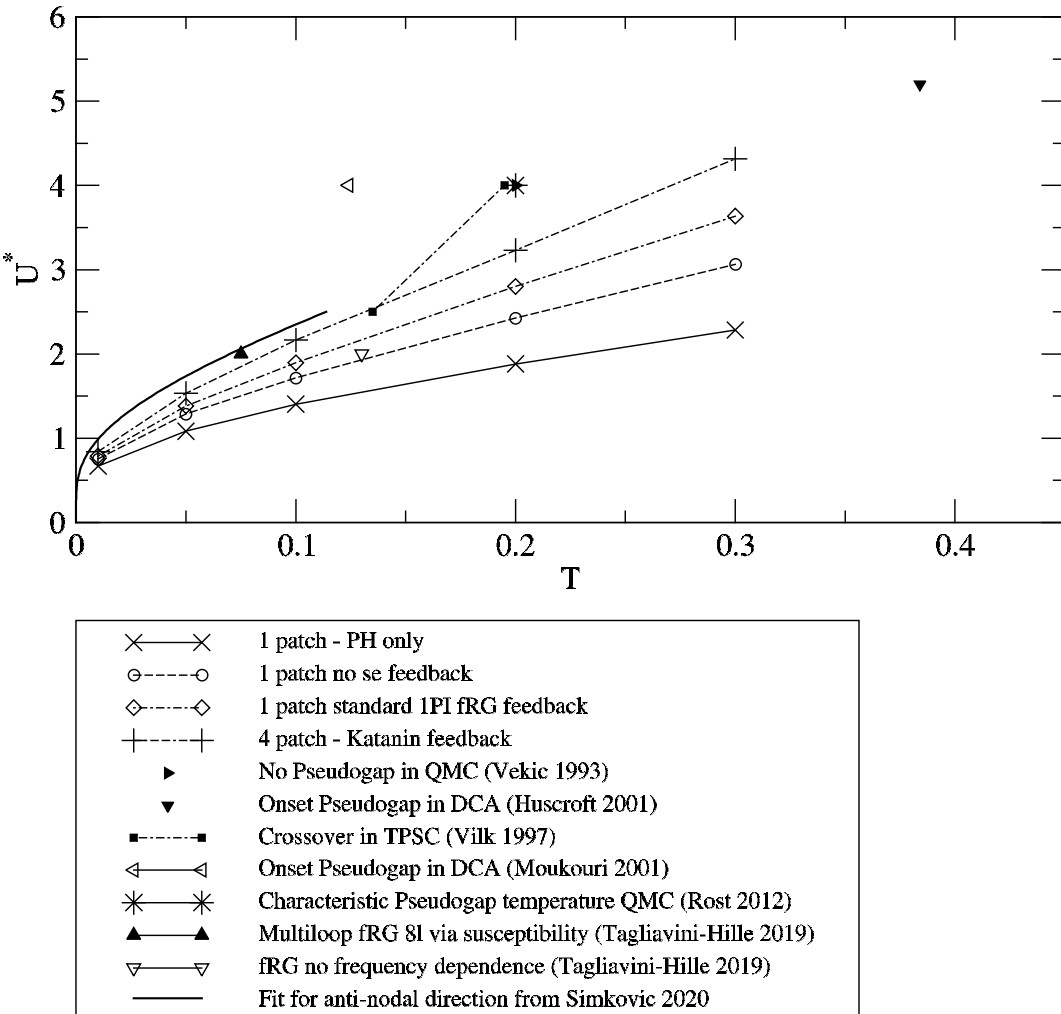

Figure 4: Comparison of the "strong-correlations" fRG scales obtained in this work with various characteristic temperature scales from other methods in the literature [14, 39–43] (Data from [39] and [43] coincide within the available accuracy). The solid line shows the fit for the cross-over at the anti-nodal direction from [52]. Given the interpretation of all these scales as smooth transitions from a normal state to an unusual state, the Katanin-corrected version of the fRG seems to agree best. "1-patch" and "4-patch" refer to the angular resolution of the momentum grid along the Fermi surface from $(\pi, 0)$ to $(\pi/2, \pi/2)$.

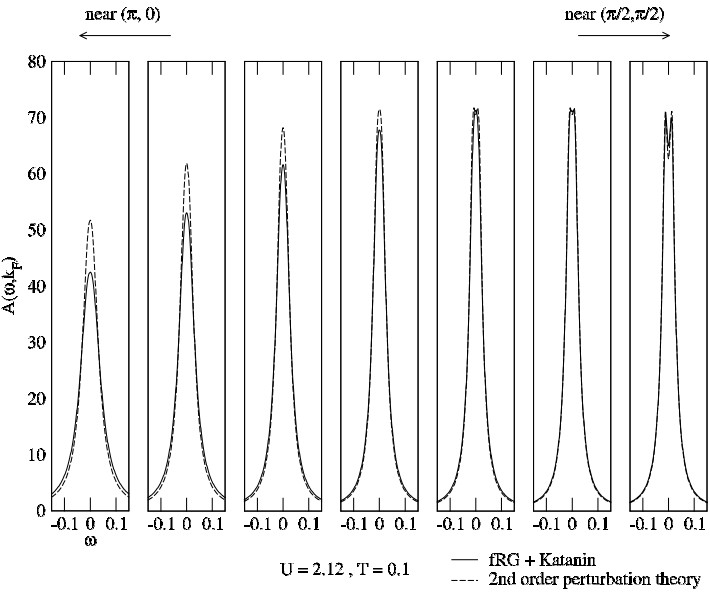

Figure 5: Spectral functions for $T = 0.1$ obtained in the Katanin-corrected version for the case of seven angular patches between the anti-nodal and nodal point on the Fermi surface, i.e. the perfectly nested Umklapp surface. Dotted lines show the result for SOPT for the same value of the bare coupling. We note that near the nodal line thermal effects dominate, since SOPT and fRG essentially fall on top of each other. Towards the anti-nodal direction the influence of the fRG flow increases, yet the picture remains rather similar to SOPT, not only qualitatively but also quantitatively.

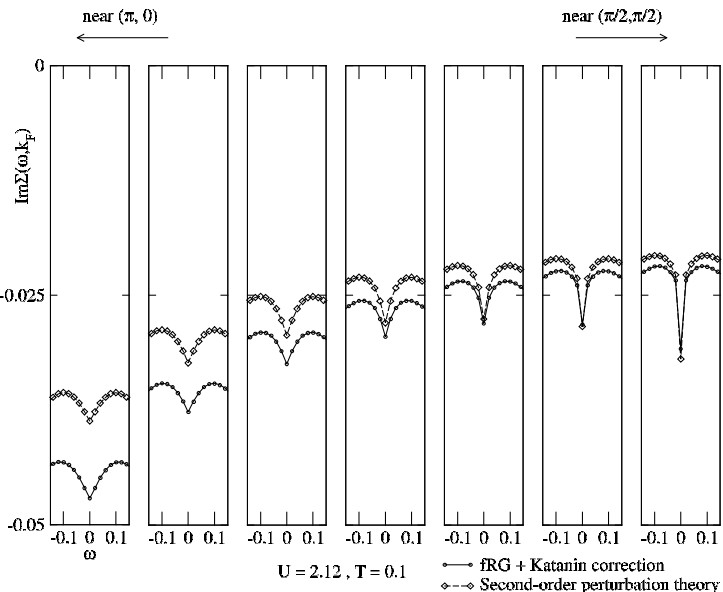

Figure 6: Self-energy corresponding to Figure 5

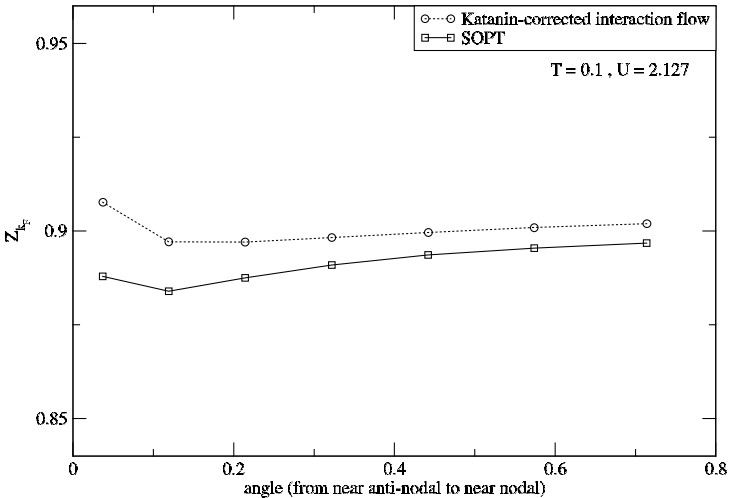

Figure 7: $Z$-factors along the Fermi surface as a function of angle for $T = 0.1$. SOPT and fRG results are nearly identical.

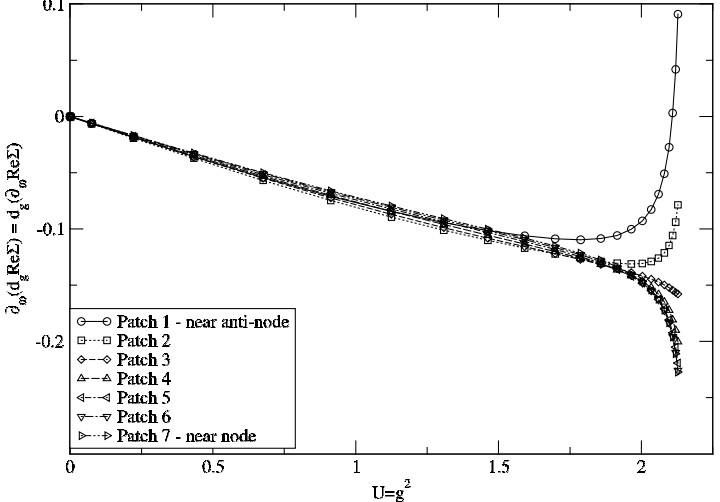

Figure 8: Flow of the "slope" of $\frac{d}{dg}Re\Sigma$ on the Fermi surface from near anti-nodal (patch 1) to near nodal (patch 7) for $T = 0.1$. The slope is defined as the Quasi-Fermi-liquid approximation, in analogy to Figure 2. The total number of discrete patches for this calculation was 728.

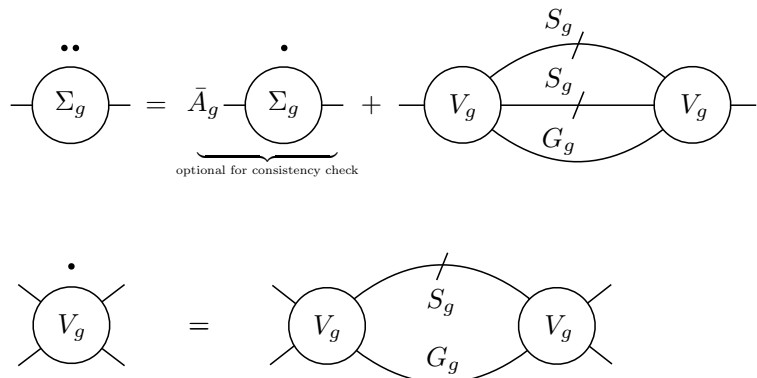

Figure 9: Graphical representation for the flow equation of the self-energy and the two-particle vertex in the 1-PI scheme.

Figure 10: Graphical representation for the flow equations after extracting the two-loop contributions by means of the second derivative of the self-energy.

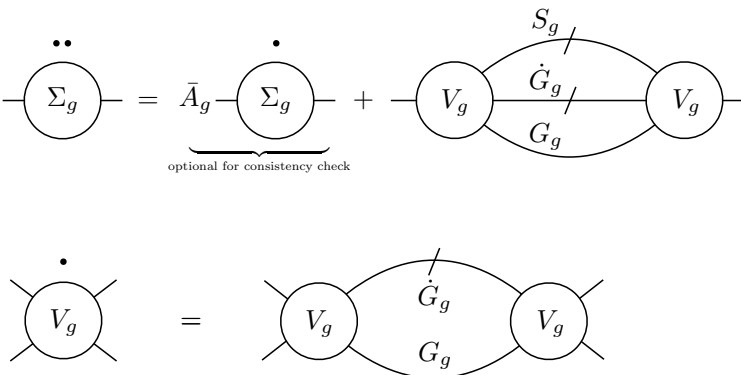

Figure 11: As Figure 10 plus the inclusion of the Katanin correction for the vertex.

## Acknowledgements

The authors gratefully acknowledge the computing time granted through JARA-HPC on the supercomputer JURECA [53] at Forschungszentrum Juelich.

## A  Details about the method

For this work we used and extended the so-called interaction-flow version of the fRG, a detailed description of which as well as explicit computations of the two-particle interaction vertex for the two-dimensional Hubbard model are given in [22]. A related scheme in the context of high-energy physics was proposed in [29]. It is defined by using one-particle irreducible (1-PI) vertex functions and choosing a homogeneous scaling factor $g$ in the bare one-particle propagator as the flow parameter. We subsequently employ the standard truncation of the hierarchy of flow equations by setting the six-point function to zero during the flow and approximating the interaction vertex by its static components.

It is important to note that in this scheme the flow parameter does not regularise the bare theory at $T = 0$. Rather, the method can only be applied at finite temperatures given that we restrict ourselves to the normal, non-symmetry-broken phase. We wish to extend the interaction flow method in order to include single-particle spectral properties. For that purpose, we calculate the retarded self-energy $\Sigma^R(\omega, \mathbf{k}) := \Sigma(\omega + i0^+, \mathbf{k})$ directly on the real-frequency axis in analogy to the route taken in [23]. It will be denoted as $\Sigma(\omega, \mathbf{k})$ throughout this work. A priori, in the 1-PI scheme the two-loop contribution, which is necessary to obtain one-particle spectral properties for a frequency-independent interaction vertex, is non-local in the flow parameter. In other words, a frequency-dependent self-energy is a result of either a reinsertion procedure or a partial integration [54–56]. In the interaction-flow method this can be avoided by shifting the flow equation to second order in the flow parameter, at the cost of introducing an additional approximation which will be outlined below.

In a spin-rotationally invariant system we can separate the spin and momentum dependence of the interaction vertex and work with functions of momenta only. Here, we use the singlet-triplet-representation as also employed in e.g. [23, 57, 58] and denote these functions as $V_g^{s,t}$ or briefly as $V_g$ where the spin-representation is implicitly understood.

For a comprehensive and general overview of the fRG framework we defer the reader to the dedicated literature such as e.g. [8] and references therein. We shall here outline the main properties and definitions only in a rather compact way and put emphasis on the additional steps and aspects that are introduced.

### A.1 Flow equations and propagators

The flow equation for the self-energy in the 1-PI scheme with flow parameter $g$ consists of a generalised tadpole diagram as shown in Figure 9 and reads in short-hand notation and the often-used level-2 truncation of the flow hierarchy [8, 54]

$$\frac{d}{dg}\Sigma_g = S_g \circ V_g = S_g \circ \int_0^g d\tilde{g} \sum V_{\tilde{g}}\, G_{\tilde{g}}\, S_{\tilde{g}}\, V_{\tilde{g}}\,, \tag{4}$$

where the sum $\sum$ denotes contractions over internal frequencies, momenta and spin and $\circ$ is a short-hand notation for the same kind of contraction of the single-scale propagator $S_g$ with the effective vertex function $V_g$ given by the integral expression.

The flow of the four-point function is given by the one-loop diagram [22] as shown in Figure 9. It reads in short-hand notation

$$\frac{d}{dg}V_g = \sum V_g\, G_g\, S_g\, V_g\,. \tag{5}$$

In the interaction-flow scheme $S_g$ and $G_g$ are given as (cf. [22] up to a sign convention)

$$
\begin{aligned}
S_g &= (i\omega - \xi_{\mathbf{k}}^0 - g\Sigma_g)^{-2}(i\omega - \xi_{\mathbf{k}}^0) \\
G_g &= g(i\omega - \xi_{\mathbf{k}}^0 - g\Sigma_g)^{-1}\,.
\end{aligned}
\tag{6}
$$

### A.2 Accessing the frequency dependence of the self-energy

As stated above, we want to extend the interaction-flow scheme to calculate the frequency-dependent self-energy without any reinsertion procedure, since the latter will become unfeasible at the resolutions we are eventually aiming at due to various numerical reasons (performance, memory, step resolution, accuracy). Yet, we also need to stick to the static approximation for the two-particle interaction when working with real frequencies. This is not possible by direct means of the one-loop diagram since it fails to yield any frequency dependence at all in this case. The strategy to overcome this restriction consists in differentiating the flow equation of the two-point function once more with respect to $g$. This explicitly extracts a two-loop contribution on the right-hand side (rhs) which is the minimal ingredient to produce a frequency dependence. In a nutshell we will do this:

$$\dot{\Sigma} \to \ddot{\Sigma} = \dot{S}_g \circ V_g + S_g \circ \dot{V}_g\,. \tag{7}$$

The second term then gives us the two-loop contribution with an explicit frequency-dependence. The first term can either be dropped since again it produces no explicit frequency dependence, or it can approximately be represented by a term proportional to

$\dot{\Sigma}$. We mainly opted for the latter since it is exact at the beginning of the flow and thus seems more accurate to us *a priori*. This ambiguity is a deficiency on the one hand, yet on the other hand it allows for internal consistency checks. For some cases we ran both variants and found that they agree up to the point at which we enter a strong-coupling regime where the fRG as a weak-coupling approximation becomes invalid anyhow. The structure of the final equations including the Katanin-correction turns out as

$$
\begin{aligned}
\frac{d}{dg}\Sigma_g &= \dot{\Sigma}_g \\
\frac{d}{dg}\dot{\Sigma}_g &= A_g \dot{\Sigma}_g + S_g \dot{V}_g \\
\frac{d}{dg}V_g &= V_g \dot{G}_g G_g V_g \,.
\end{aligned}
\tag{8}
$$

We now outline the three steps we took to arrive at this set of flow equations which are local in the flow parameter, allow for the calculation of a frequency-dependent self-energy, and include the Katanin-correction.

### A.3  Step 1: Two-loop contribution local in the flow parameter without incorporating any self-energy feedback

When neglecting self-energy feedback in the interaction-flow method we have $S_g = (i\omega - \xi_{\mathbf{k}}^0)^{-1}$ and $G_g = g(i\omega - \xi_{\mathbf{k}}^0)^{-1}$ [22]. Thus, in this approximation $S_g$ is independent of the flow parameter and in fact given by the bare propagator of the non-interacting system. An additional differentiation of equation (4) with respect to $g$ then yields

$$
\frac{d^2}{dg^2}\Sigma_g = S_g \circ \sum V_g G_g S_g V_g = g \sum V_g G^0 G^0 G^0 V_g \,,
\tag{9}
$$

since $\frac{d}{dg}S_g$ vanishes and where again all self-energy feedback has been neglected in the final expression. In the same manner the flow-equation for the interaction vertex takes the standard form

$$
\frac{d}{dg}V_g = g \sum V_g G^0 G^0 V_g \,.
\tag{10}
$$

We thus arrive at flow equations for the self-energy and the four-point function which are *purely local* in $g$, the virtue of this being the fact that they can now be treated in a very similar way as in [23] and we need to evaluate the two-loop contribution only locally in $g$ without resorting to a reinsertion procedure. This is of major importance since it reduces the numerical effort as well as the memory requirements by at least an order of magnitude compared to a reinsertion algorithm. In particular, we will profit from this in calculations where the Katanin-correction is included, which is the main step forward in terms of further developing the quantitative applicability of the fRG method as such.[3]

We shall make a few comments at this point to relate the set-up to other schemes:

---

[3]A partial integration as done in [54] is not applicable here since it would force us to fix the final value of the flow parameter *a priori* which would in turn completely spoil the advantage of the interaction flow.

### A.3.1 Comment 1: On the static - or rather instantaneous - approximation

Throughout this work we approximate the interaction vertex by setting all frequencies to zero during the flow, thereby replacing it by its static components and projecting values at finite frequency onto the static sector. (Taking a function to be constant in frequency really translates into an instantaneous functional form in time, but we DO project on the static sector.) This is more unfavourable in the interaction-flow then e.g. in a Wick-ordered scheme as employed in [23], since the support for the propagators on the rhs of the flow equation does not shrink around the (non-interacting) Fermi surface. On the contrary, the support is always given by the complete Brillouin zone and neglecting the frequency dependence will introduce a larger uncertainty. In turn, certain contributions stemming from scattering processes in the forward channel of particle-hole contributions are underestimated in schemes with sharp energy cutoffs since their contribution stems from a narrow region around the Fermi surface, the region being increasingly narrow with decreasing temperature. These contributions are treated on equal footing with all other contributions in the interaction flow [22]. We find that the use of a purely static vertex function in combination with the interaction-flow method captures pseudogap phenomena in the single-particle spectral function only via emerging effects in the derivative of observables. We first considered this to be at least partly due to the lack of any frequency dependence of the interaction vertex. However, similar effects are also prominent in fRG schemes which do include such an explicit frequency dependence. It turns out that a subsequent additional improvement by means of implementing the Schwinger-Dyson equation lifts the effect up into the observables themselves [48].

### A.3.2 Comment 2: Fermi surface shifts

The leading order effect of the flowing self-energy is a shift, or rather a deformation, of the Fermi surface. If this deformation became large during the flow this would signal a qualitative breakdown of the approximation. In previous works it was found that the quantitative effect of these deformations can be expected to be rather small in a reasonably large regime of the flow [59], which is consistent with previous findings in which self-consistent FLEX calculations were done to investigate the same effect [60]. However these results have been obtained for a restricted set of parameters, namely filling/chemical potential, $t'$ and $U$. In this work this can be ignored since we work at half filling and perfect nesting, and thus there is no such shift due to symmetry reasons. Yet, when moving away from half filling and/or introducing a finite $t'$, we can conduct approximate calculations of the effective $t'$ at the end of each flow in order to check whether the deformation remains small.

### A.4 Step 2: Inclusion of self-energy feedback via the Quasi-particle weight ($Z$-factor)

The main objective of this work is to implement self-energy feedback in the numerical treatment of the flow equations within the interaction-flow method in a simple and efficient way. The two quantities through which the self-energy enters the flow equation are the single-scale propagator

$$S_g = (i\omega - \xi_{\mathbf{k}}^0 - g\Sigma_g)^{-2}(i\omega - \xi_{\mathbf{k}}^0),\tag{11}$$

and the scale-dependent full propagator

$$G_g = g(i\omega - \xi_{\mathbf{k}}^0 - g\Sigma_g)^{-1} = g\,G(g\Sigma_g) = g\,G(\Sigma(g^2 U)),\tag{12}$$

which both appear as internal lines in the diagrams in Figure 10. With the last relation for the dressed propagator $G_g$ we recall that the quantity $g\Sigma_g$ at scale $g$ in the interaction flow also

constitutes the final self-energy for the bare interaction $g^2 U$ [22]. To introduce approximations for these internal propagators we apply the standard notion of the quasi-particle weight and introduce the $Z$-factor as

$$G_g = g\, G(g\Sigma_g) \approx g\, Z_g \,(i\omega - \xi_{\mathbf{k}}^0)^{-1}\,, \tag{13}$$

where

$$Z_g = Z(g\,\Sigma_g) := (1 - \partial_\omega \mathrm{Re}(g\Sigma_g(\omega,\mathbf{k}))|_{\omega=\xi_{\mathbf{k}}^0})^{-1}\,, \tag{14}$$

in its standard definition. Here we have neglected Fermi surface shifts, band structure renormalisation and damping. Note that in numerical practice we always work on the real axis and use the explicit definitions on the Matsubara axis for convenience only, i.e. the replacement $i\omega \to \omega + i0^+$ is to be implicitly understood.

$S_g$ is approximated analogously and thus we will work with the following approximations for $G_g$ and $S_g$:

$$\begin{aligned} S_g &= Z_g^2 \,(i\omega - \xi_{\mathbf{k}}^0)^{-1} \\ G_g &= g\, Z_g \,(i\omega - \xi_{\mathbf{k}}^0)^{-1}\,. \end{aligned} \tag{15}$$

With this we can include effects of the self-energy feedback in the flow-equations which stem from the flow of the quasi-particle weight function $Z_g$. Via (15) we have implicitly compensated for the leading singularity arising from Fermi-surface shifts by using the bare dispersion during the flow, thereby keeping the FS fixed, even away from half filling and/or perfect nesting.[4] The next-leading corrections to the self-energy within a Fermi-liquid-like description are the quasi-particle weight, the scattering amplitude or inverse lifetime, and the change of the band structure. The latter two have both been omitted in (15) as we focus on the effect of the quasi-particle weight. Treating life-time effects and/or band-structure renormalisation in addition would render the computational task much more involved and can be envisaged for future extensions. At the technical level we calculate the self-energy on the real-frequency axis at each stage of the flow, determine from this the $Z$-factor and insert it in the flow equations. Note that in general the $Z$-factor is k-dependent, along the Fermi surface as well as perpendicular to it. Here we project $Z_{\mathbf{k}}$ onto the corresponding patch $Z_{\mathbf{k}_F}$, on the Fermi surface. The insertion of $Z$-factors on internal lines is straight forward for the flow equation of the interaction vertex. For the flow equation of the self-energy it involves some more steps which we will outline in the following.

The flow equation for the interaction vertex is readily generalized by inserting (15) into (5) and reads in the usual short-hand notation

$$\frac{d}{dg} V_g = g \sum V_g \, Z_g^2 \, G_g^0 \, Z_g \, G_g^0 \, V_g\,. \tag{16}$$

In order to deduce the flow equation for the self-energy we first revert to equation (4) given by

$$\frac{d^2}{dg^2} \Sigma_g = S_g \circ \int_0^g d\tilde{g} \sum V_{\tilde{g}} \, G_{\tilde{g}} \, S_{\tilde{g}} \, V_{\tilde{g}}\,. \tag{17}$$

---

[4]Away from half filling and/or perfect nesting this procedure needs to be verified at the end of each calculation by checking the changes to the effective hopping parameter t', as mentioned above.

As in the case without self-energy feedback we take the second derivative with respect to $g$, yielding

$$\frac{d^2}{dg^2}\Sigma_g = (\frac{d}{dg}S_g) \circ \int_0^g d\tilde{g} \sum V_{\tilde{g}} G_{\tilde{g}} S_{\tilde{g}} V_{\tilde{g}} \quad + \quad \sum V_g G_g S_g S_g V_g. \tag{18}$$

Since we use a frequency-independent two-particle vertex, we could at this stage again drop the one-loop term, since it does not contribute to the frequency dependence of the self-energy and thus does not contribute to $Z_g$. In practice, this serves as a cross-check only. We have the option to do better as we will see, and since $\frac{d}{dg}S_g$ does not vanish anymore, the one-loop term should be dealt with in an approximate manner. This is the price we have to pay on our way to an rhs which at the end will be local in $g$.

From equation (15) we have

$$\begin{aligned}
\frac{d}{dg}S_g &= 2(\frac{d}{dg}Z_g)Z_g\,(i\omega - \xi_{\mathbf{k}}^0)^{-1} \\
&= 2(\frac{d}{dg}Z_g)Z_g^{-1}Z_g^2(i\omega - \xi_{\mathbf{k}}^0)^{-1} \\
&= 2(\frac{d}{dg}Z_g)Z_g^{-1}S_g \\
&= A_g\,S_g\,,
\end{aligned} \tag{19}$$

where $A_g := 2(\frac{d}{dg}Z_g)Z_g^{-1}$. At this stage we have lost all the advantage of a local rhs since the first term on the rhs of equation (18) makes the non-local integral expression reappear, since $\frac{d}{dg}S_g$ appears inside the contraction with the interaction vertex and thus runs over *internal* momenta. However, *if* $A_g$ was independent of $k$ we could pull the prefactor $2(\frac{d}{dg}Z_g)Z_g^{-1}$ out of the contraction and write

$$\begin{aligned}
(\frac{d}{dg}S_g) \circ \int_0^g d\tilde{g} \sum V_{\tilde{g}} G_{\tilde{g}} S_{\tilde{g}} V_{\tilde{g}} &= (2(\frac{d}{dg}Z_g)Z_g^{-1})\left(S_g \circ \int_0^g d\tilde{g} \sum V_{\tilde{g}} G_{\tilde{g}} S_{\tilde{g}} V_{\tilde{g}}\right) \\
&= A_g \dot{\Sigma}_g\,,
\end{aligned} \tag{20}$$

where we have inserted (4). We could thereby replace the numerically infeasible non-local integral expression by a simple feedback of $\dot{\Sigma}$ on $\ddot{\Sigma}$ which induces zero numerical costs and keeps the rhs local in $g$.

We now recall that we project the $Z$-factor onto the (non-interacting) Fermi surface and thus this procedure is valid in a strictly isotropic system without any symmetry breaking. Here, however, we deal in general with non-isotropic systems and in order to make use of this simplification we are forced to introduce an additional approximation: We replace $A_g$ by its average along the Fermi surface patches and denote this as $\bar{A}_g$. We then arrive at the following expression which we can treat numerically:

$$\frac{d^2}{dg^2}\Sigma_g = \bar{A}_g\frac{d}{dg}\Sigma_g \quad + \quad \sum V_g G_g S_g S_g V_g. \tag{21}$$

In practice, we found that the results do not vary considerably up to the point when the flow starts to leave the weak-coupling region. In a transient zone $A_g$ actually does develop an

angular dependence via $\frac{d}{dg}Z_g$ as seen in Figure 8, which indicates the tendency towards pseudogap formation near the anti-nodes. By using the average we suppress the internal feedback of this tendency on the flow. It would be desirable to include it, but this is numerically not feasible at this stage.

Since we compute $Z_g$ from $g\Sigma_g$ in the numerical implementation we rephrase $\frac{d}{dg}Z_g$ starting from equation (14) as

$$
\begin{aligned}
\frac{d}{dg}Z_g &= \frac{d}{dg}(1 - \partial_\omega \mathrm{Re}(g\Sigma_g(\omega,\mathbf{k}))|_{\omega=\xi_\mathbf{k}^0})^{-1} \\
&= (1 - \partial_\omega \mathrm{Re}(g\Sigma_g(\omega,\mathbf{k}))|_{\omega=\xi_\mathbf{k}^0})^{-2} \frac{d}{dg}(\partial_\omega \mathrm{Re}(g\Sigma_g(\omega,\mathbf{k}))|_{\omega=\xi_\mathbf{k}}) \\
&= Z_g^2 \frac{d}{dg}(\partial_\omega \mathrm{Re}(g\Sigma_g(\omega,\mathbf{k}))|_{\omega=\xi_\mathbf{k}^0}) \\
&= Z_g^2 \{\partial_\omega \mathrm{Re}(\Sigma_g(\omega,\mathbf{k}))|_{\omega=\xi_\mathbf{k}^0} + \partial_\omega \mathrm{Re}(g\frac{d}{dg}\Sigma_g(\omega,\mathbf{k}))|_{\omega=\xi_\mathbf{k}^0}\},
\end{aligned}
\tag{22}
$$

and obtain via (19)

$$
\frac{d}{dg}S_g = 2Z_g\{\partial_\omega \mathrm{Re}(\Sigma_g(\omega,\mathbf{k}))|_{\omega=\xi_\mathbf{k}^0} + g\partial_\omega \mathrm{Re}(\frac{d}{dg}\Sigma_g(\omega,\mathbf{k}))|_{\omega=\xi_\mathbf{k}^0}\}S_g = A_g S_g.
\tag{23}
$$

Finally we also replace $S_g$ and $G_g$ in the two-loop term term according to (15), thus using

$$
\sum V_g G_g S_g S_g V_g \approx \sum V_g Z_g G_g^0 Z_g^2 G_g^0 Z_g^2 G_g^0 V_g.
\tag{24}
$$

This completes the set of truncated flow equations for the case of the standard, non-Katanin-corrected self-energy feedback within the 1-PI fRG equations. We shall again offer a comment and argue that attention needs to be paid to certain aspects.

### A.4.1 Comment 3: Damping and the definition of the quasi-particle weight

There remains, as noted above, an ambiguity concerning the usage of $Z$ in the numerical algorithm caused by neglecting the scattering, i.e. $\mathrm{Im}\Sigma(\omega = \epsilon_\mathbf{k})$. Formally, already at the perturbative level there exists a high-energy region in momentum space for energies sufficiently far away from the Fermi surface where the $Z$-factor by definition is larger than one.[5] However, this is in regions where also the scattering rate is very high. When self-energy feedback is neglected this issue does not arise, but when we do include it via the $Z$-factor we need to deal with this. We chose to project the $Z$-factor onto the corresponding patch momentum on the Fermi surface since we found that the slope of the real part of the self-energy does not change significantly around a substantial energy region around the Fermi surface. Regions further away from the Fermi surface should not change the results, at least not qualitatively, by means of usual suppression arguments, and also since large scattering should suppress these contributions further. Yet, a more elaborate way of implementing the weight of high-energy modes in the numerical treatment is desirable, in order to verify this procedure.

---

[5]We note that Fermi-liquid theory is strictly speaking only defined *after* the high-energy degrees of freedom have been integrated out, and the effective description is restricted to a narrow region around the Fermi surface. Here, we may say that we "borrow" from this concept.

## A.5 Step 3: Inclusion of the Katanin-correction

We finally extend the calculation to include the Katanin-correction, i.e. we replace $S_g$ by the full derivative of $G_g$ with respect to $g$ in the flow equation of the two-particle interaction (only!) [21]. The corresponding term which we need to include in our approximation is straight-forwardly computed from (13) and (14) by first evaluating

$$
\begin{aligned}
\frac{d}{dg} Z_g &= (1 - \partial_\omega \mathrm{Re}(g\Sigma_g(\omega, \mathbf{k}))|_{\omega=\xi_\mathbf{k}^0})^{-2} \frac{d}{dg} \partial_\omega \mathrm{Re}(g\Sigma_g(\omega, \mathbf{k}))|_{\omega=\xi_\mathbf{k}^0} \\
&= Z_g^2 \partial_\omega \mathrm{Re}(\Sigma_g(\omega, \mathbf{k}))|_{\omega=\xi_\mathbf{k}^0} + Z_g^2 \partial_\omega \mathrm{Re}(g\frac{d}{dg}\Sigma_g(\omega, \mathbf{k}))|_{\omega=\xi_\mathbf{k}^0},
\end{aligned}
\tag{25}
$$

and using this to obtain

$$
\begin{aligned}
\frac{d}{dg} G_g &= Z_g^2 (1 + g^2 \partial_\omega \mathrm{Re}(\frac{d}{dg}\Sigma_g(\omega, \mathbf{k}))|_{\omega=\xi_\mathbf{k}^0})(i\omega - \xi_\mathbf{k}^0)^{-1} \\
&= K_g S_g,
\end{aligned}
\tag{26}
$$

where we define

$$
K_g := (1 + g^2 \partial_\omega \mathrm{Re}(\frac{d}{dg}\Sigma_g(\omega, \mathbf{k}))|_{\omega=\xi_\mathbf{k}^0}).
\tag{27}
$$

We refer to $K_g$ as the Katanin-factor. This expression should cause immediate attention since it is *a priori* possible for $K_g$ to become negative during the flow, which in turn can lead to a change from an *increase* to a *decrease* (and vice-versa) in the respective contributions to the flow of certain channels in the interaction vertex at some scale $g^*$. Most importantly, $K_g$ may rapidly approach zero at a scale $g^*$. This in turn can lead to a saturation of all vertex functions and cure the ladder-like divergences which are present without the Katanin correction. In practice, we observed such a saturation for coarse-grained discretisations of the Brillouin zone, but with increasing resolution, i.e. increasing number of patches used in momentum space, the two-particle interaction diverges before such an effect sets in. Yet, we consider this to be the origin of the dependence of the results in Figure 4 on the discretisation for the Katanin-corrected scheme. In a broader sense we may speak of this as a "finite-size-like" effect in momentum space.

When introducing the Katanin correction, some of our arguments concerning upper and lower bounds in the next section as well as some general features of the flow become some-what more complicated, since $K_g$ may change sign around a certain scale $g^*$ which invalidates some of our arguments used above. In practice, we always verify *a posteriori* that our assumptions are actually valid.

The Katanin-correction is readily implemented by using equation (26) and inserting the resulting expression for $\frac{d}{dg}G_g$ in the flow equation for the interaction vertex. Note that this also means replacing the formerly "inner" $S_g$ by $K_g S_g$ in the two-loop diagram of the self-energy. The flow equation for the interaction then reads

$$
\frac{d}{dg} V_g = g \sum V_g K_g Z_g^2 G_g^0 Z_g G_g^0 V_g.
\tag{28}
$$

For the self-energy the two-loop term is replaced accordingly by

$$
\sum V_g G_g S_g S_g V_g \rightarrow \sum V_g Z_g G_g^0 Z_g^2 G_g^0 K_g Z_g^2 G_g^0 V_g.
\tag{29}
$$

We note that inserting the inverse relations of equations (14) and (27) simplifies $A_g$ in equation 19 to

$$A_g = -\frac{2}{g}\left(1 - Z_g K_g\right).\tag{30}$$

## A.6 Alternative derivation of $S_g$

We shall mention that a slightly different route leads to the same expression for $S_g$. This provides an fRG-internal consistency check as follows: If we instead start from the general relation [8]

$$S_g = \frac{d}{dg} G_g \mid_{\Sigma_g fixed},\tag{31}$$

and use the approximation of $G_g$ in (15), this leads to

$$S_g = Z_g \left(i\omega - \xi_{\mathbf{k}}^0\right)^{-1} + g\left(i\omega - \xi_{\mathbf{k}}^0\right)^{-1}\frac{d}{dg} Z_g \mid_{\Sigma_g fixed},\tag{32}$$

and with

$$\frac{d}{dg} Z_g \mid_{\Sigma_g fixed} = \left(1 - \partial_\omega \mathrm{Re}(g\Sigma_g(\omega,\mathbf{k}))\mid_{\omega=\xi_{\mathbf{k}}}\right)^{-2}\frac{d}{dg}\partial_\omega \mathrm{Re}(g\Sigma_g(\omega,\mathbf{k}))\mid_{\omega=\xi_{\mathbf{k}}} \mid_{\Sigma_g fixed}\tag{33}$$

$$= Z_g^2 \partial_\omega \mathrm{Re}(\Sigma_g(\omega,\mathbf{k}))\qquad \left(\Sigma_g \text{ is held fixed, thus we have to take }\frac{d}{dg}\Sigma_g = 0\right),\tag{34}$$

we get

$$S_g = \left(Z_g + Z_g^2 \partial_\omega \mathrm{Re}(g\,\Sigma_g(\omega,\mathbf{k}))\right)\left(i\omega - \xi_{\mathbf{k}}^0\right)^{-1}.\tag{35}$$

Inverting equation (14) and inserting this into equation (35) yields the same equation for $S_g$ as in (15).

## A.7 Alternative derivation of $\partial_g S_g$

We note that we introduced the simplified expression from (15) for $S_g$ first and then took the derivative with respect to $g$, which could in principle lead to an oversimplification. Starting instead from (11) we have

$$\begin{aligned}\partial_g S_g &= 2\left(i\omega - \xi_{\mathbf{k}}^0 - g\Sigma_g\right)^{-3}\partial_g(g\,\Sigma_g)\left(i\omega - \xi_{\mathbf{k}}^0\right)\\&= 2\left(i\omega - \xi_{\mathbf{k}}^0 - g\Sigma_g\right)^{-1}\partial_g(g\,\Sigma_g)S_g,\end{aligned}\tag{36}$$

and for consistency we need to at least motivate the relation

$$(\partial_g Z_g)Z_g^{-1} = \left(i\omega - \xi_{\mathbf{k}}^0 - g\Sigma_g\right)^{-1}\partial_g(g\,\Sigma_g).\tag{37}$$

In fact on the formal level we can write

$$
\begin{aligned}
(i\omega - \xi_{\mathbf{k}}^0 - g\Sigma_g)^{-1} \partial_g(g\,\Sigma_g) &= -\partial_g \ln(i\omega - \xi_{\mathbf{k}}^0 - g\Sigma_g) \\
&= \partial_g \ln((i\omega - \xi_{\mathbf{k}}^0 - g\Sigma_g)^{-1}) \\
&\approx \partial_g \ln(Z_g (i\omega - \xi_{\mathbf{k}}^0)^{-1}) \\
&= \partial_g (\ln(Z_g) - \ln(i\omega - \xi_{\mathbf{k}}^0)) \\
&= \partial_g \ln(Z_g) \\
&= (\partial_g Z_g) Z_g^{-1} \quad \text{q.e.d.,}
\end{aligned}
\tag{38}
$$

where we have inserted equation (15) in the third step. This provides a consistency check to some extent.

## B   Estimating the impact of using $\bar{A}_g$

Here we wish to offer some additional thoughts on the approximation for the one-loop contribution in equation(21). While we cannot make an exact replacement as suggested in equation (20), we can still use it to come up with some estimates/approximations concerning the first term on the rhs of equation (18) by finding upper and lower bounds, allowing us to compute approximate solutions and estimate their quality.

In practice, we calculate the flow of the imaginary part of the self-energy $\mathrm{Im}\Sigma_g$ and obtain the real part at any scale $g$ via Kramers-Kronig relations. Defining $A_g^{min,max} := min, max\{A_g\}$ over all patches in momentum space, anticipating that $A_g < 0$ and that $\mathrm{Im}\partial_g \Sigma_g < 0\,\forall g$ in the retarded limit which we use, we may expect the following to hold:

$$
A_g^{max}\, \partial_g \mathrm{Im}\Sigma_g + \mathrm{Im}\sum V_g\, G_g\, S_g\, S_g\, V_g \;\leq\; \frac{\partial^2}{\partial g^2}\mathrm{Im}\Sigma_g \;\leq\; A_g^{min}\, \partial_g \mathrm{Im}\Sigma_g + \mathrm{Im}\sum V_g\, G_g\, S_g\, S_g\, V_g\,.
\tag{39}
$$

However, since this argument is based *only* on the lower and upper values of $A_g$ it is incomplete and not necessarily valid *a priori*, since the flow of $\partial_g \mathrm{Im}\Sigma_g$, $Z_g$ and $V_g$ also depends on whether we use $A_g^{max}$ or $A_g^{min}$. Yet, by using either replacement and comparing the numerical results we can *a posteriori* verify if this inequality is valid for all $g$ and by how much the two limits differ at each scale.

In order to actually determine $A_g$ and from this $A_g^{min,max}$ we need to numerically calculate not only $\partial_\omega \mathrm{Re}(g\Sigma_g(\omega, \mathbf{k}))|_{\omega=\xi_{\mathbf{k}}}$ during the flow, but also the corresponding expression for $\partial_g \Sigma_g$. With this double inequality we can then follow the error we make when we simplify the first term on the rhs of equation (18) when replacing it by a term proportional to $\partial_g \mathrm{Im}\Sigma_g$.

We now note that

- At the beginning of the flow we have $g = 0$, $\Sigma_g = 0$ and $\partial_g \Sigma_g = 0$. Thus by continuity arguments $A_0$ vanishes and stays small for small $g$.

- During the flow there will be a momentum-dependent build up of $\partial_\omega \mathrm{Re}(\Sigma_g(\omega, \mathbf{k}))|_{\omega=\xi_{\mathbf{k}}}$ and $g\partial_\omega \mathrm{Re}(\partial_g \Sigma_g(\omega, \mathbf{k}))|_{\omega=\xi_{\mathbf{k}}}$ which will both be negative on and near the Fermi surface in the case of ordinary behavior in the Fermi liquid sense.
  In high energy regions far away from the Fermi surface the approximation (15) is invalid due to high damping and positive slope of $\mathrm{Re}\Sigma$.

Hence we project the quantities related to derivatives of the self-energy onto the Fermi surface to avoid complications (cf. discussion below). In order to check the validity of this procedure one needs to introduce a patching scheme with a very fine grained resolution in the energy direction perpendicular to the Fermi surface. Such a calculation is planned for the future.

- While we cannot exactly eliminate the integral expression in (18) since $A_g$ is momentum-dependent, we can still make use of the general idea and insert the minimal and maximal values of $A_g$ and numerically follow upper and lower bounds, thereby estimating the quality of the approximation.

In practice, we observed that for the variations of $A_g$ which we encounter the results do not strongly depend on the choice of $A_g^{max}$ or $A_g^{min}$. This becomes invalid when the derivative of $Z$ changes sign near the anti-node, but then this happens only at the very end of the flow, when observables are rather robust. As stated in the main body: As long as the method is strictly valid, the physics stays "boring". We then see the emergence of "non-boring" effects in a region where the method becomes more and more approximate, before entering a region where we begin to see "very interesting stuff", but where the method clearly lacks reliability. We feel it is important to reiterate this statement. The fRG has certain merits and certain drawbacks, which we need to consider when interpreting the results and drawing conclusions.

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
