# Peer review of "Quasi-particle functional Renormalisation Group calculations in the two-dimensional half-filled Hubbard model at finite temperatures"

_SciPost Physics, doi:SciPost Phys. 9, 084 (2020)_

## Round 1 · Referee Report · Walter Metzner (Referee 1) · 2020-5-27

Report

The authors present a one-loop functional renormalization group (fRG) study of the two-dimensional half-filled Hubbard model at finite temperature, with a focus on the frequency-dependence of the self-energy. Using a static approximation for the two-particle vertex, the flow of the self-energy can be computed directly on the real frequency axis. While numerous fRG studies of the Hubbard model have already been published, there are only few results for the real-frequency self-energy.

The authors observe an unconventional dip of the imaginary part of the self-energy around zero frequency, which is present only at finite temperature.This feature was discovered previously within plain second order perturbation theory (as the authors correctly point out), but it didn't receive the attention it deserves. They thoroughly discuss the consequences of the dip for the spectral function and Fermi liquid behavior. Signatures of pseudogap behavior are also discussed.

I recommend publication after the points under ''Requested changes'' have been taken into account.

Requested changes

I see several points that should or could be improved, which I divide in ''mandatory'' and ''optional''.

Mandatory:

1) The thermal dip in the self-energy was observed already by Katanin and Kampf in their fRG study (Refs. 31 and 32). Their work and a comparison to their results should be discussed more thoroughly.

2) In 1.2 it is claimed that Ref. 16 is the only fRG implementation where the self-energy is computed directly on the real axis. This may be true for studies of the 2D Hubbard model, but certainly not in general. Hence, the authors should restrict the statement to the two-dimensional Hubbard model.

3) In the introduction the authors write that ''qualitatively there are hardly any issues'' (for the 2D Hubbard model). This is not true and should therefore be removed. In particular, none of the existing fRG versions is able to capture the pseudogap, and no method at all is able to deal with strange metal transport behavior.

4) The authors should not try to sell a one-loop truncation with a static vertex as a step toward ''quantitative maturity'' (in introduction), as it is clear that higher loops and frequency dependencies of the vertex change results a lot already at moderate interaction strength. The partial agreement with results from more accurate methods might be just accidental.

Optional:

1) I strongly encourage the authors to include results away from half-filling in the present paper, instead of postponing this to a future publication. This would allow one to assess to what extent the unusual self-energy dip is associated with nesting and the van Hove singularity. It shouldn't be hard to produce the data and add a few figures. The method remains the same, so that the paper would not become much longer.

Minor points and typos:

1) There are words missing in the last sentence of the first paragraph on page 3.

2) Why do the authors refer to their results in Fig. 1 as ``raw data''? What does it mean?

3) There is something missing in the condition ''omega = ...'' in line 11 on page 6. As written, the equation doesn't make sense.

4) ''FEA'' is an unusual acronym for the fluctuation-exchange approximation. Why not ''FLEX'', which is the standard one?

5) I don't see the QMC data points from Rost 2012 in Fig. 4. Were they forgotten?

  • validity: high
  • significance: high
  • originality: good
  • clarity: good
  • formatting: good
  • grammar: excellent

Author:  Daniel Rohe  on 2020-06-08  [id 852]

(in reply to Report 1 by Walter Metzner on 2020-05-27)

We thank the referee for his comments and suggestions. We agree with most of them, while for some of them we are not sure whether we fully understand the background. Please find our answers below.

Mandatory:

1) The thermal dip in the self-energy was observed already by Katanin and Kampf in their fRG study (Refs. 31 and 32). Their work and a comparison to their results should be discussed more thoroughly.

We believe there is a misunderstanding. What we termed "thermal dip" and what is e.g. shown in our Fig. 1 is not present in our view in [31,32] (we also note that these references do not address the nested half-filled case of the Hubbard model). In SOPT, there is this thermal break-down path, which however fades away at lower temperatures. What may be more relevant is the breakdown of the Fermi liquid due to correlation effects, that is advocated in Refs 31, 32 and also seen near the anti-nodal point in our data. In the revised manuscript we discuss and discriminate these two effects more clearly.

2) In 1.2 it is claimed that Ref. 16 is the only fRG implementation where the self-energy is computed directly on the real axis. This may be true for studies of the 2D Hubbard model, but certainly not in general. Hence, the authors should restrict the statement to the two-dimensional Hubbard model.

We agree. We have softened our statement in the manuscript. Further, we added two works by Förchinger on general fRG formalism and by Enss with a real-frequency self-energy analysis in other systems. Beyond this, we would also be grateful to get hints on concrete references where this has been done since we are not aware of these works and have not been able to find them.

3) In the introduction the authors write that ''qualitatively there are hardly any issues'' (for the 2D Hubbard model). This is not true and should therefore be removed. In particular, none of the existing fRG versions is able to capture the pseudogap, and no method at all is able to deal with strange metal transport behavior.

We were imprecise and will improve on this section. We now write 'Qualitatively, the method has been able to provide relevant insights, but after more than 20 years the goal now is to also reach a better definition of its quantitative character.' What we meant to say was that the methods qualitatively agree amongst each other in most respects. Of course there are effects that are captured by none of them. Yet, some fRG-works do capture pseudogap-like features or the onset thereof, e.g. references [31,32] and [16].

4) The authors should not try to sell a one-loop truncation with a static vertex as a step toward ''quantitative maturity'' (in introduction), as it is clear that higher loops and frequency dependencies of the vertex change results a lot already at moderate interaction strength. The partial agreement with results from more accurate methods might be just accidental.

Again, we were imprecise and changed the wording, no longer using 'maturity', see answer to 3) above. The line of thinking in our motivation was to include aspects which have shown to improve fRG results quantitatively, in particular the feedback of the self-energy in the Katanin-corrected version (reference [14]). Of course there are effects that remain out of scope such as higher-loop corrections, and also the fact that we use a static vertex together with a frequency-dependent self-energy violates the general line of though in reference [14]. That is the reason why we did calculations for both cases, with and without Katanin-correction, to compare them to results from other methods. Whether agreement is accidental or not is a matter of discussion, of course. Here, we present data, no more no less.

Optional:

1) I strongly encourage the authors to include results away from half-filling in the present paper, instead of postponing this to a future publication. This would allow one to assess to what extent the unusual self-energy dip is associated with nesting and the van Hove singularity. It shouldn't be hard to produce the data and add a few figures. The method remains the same, so that the paper would not become much longer.

The computation does require substantial compute time. We can include a few results for these cases, but a comprehensive study is out of scope for this work. We would prefer to disentangle the peculiar case of the half-filled perfectly nested case from the general, non-half filled non-nested cases to avoid a loss in oversight.

Minor points and typos:

1) There are words missing in the last sentence of the first paragraph on page 3.

Confirmed. We repaired this in the revised manuscript.

2) Why do the authors refer to their results in Fig. 1 as raw data''? What does it mean?

We were incomplete here: "Raw" means these are the frequencies at which the self-energy is actually computed. To calculate the real part, the Z-factor and the spectral function we need to compute a Kramers-Kronig-transform. This requires a much finer resolution of the data, which is at the numerical level achieved by an Akima-spline interpolation. The result of the latter is the "post-processed" data as contrasted to the "raw" data. We make this clearer now in the revised version.

3) There is something missing in the condition ''omega = ...'' in line 11 on page 6. As written, the equation doesn't make sense.

Confirmed. '\Sigma' is missing. Has been fixed.

4) ''FEA'' is an unusual acronym for the fluctuation-exchange approximation. Why not ''FLEX'', which is the standard one?

The authors of that very reference used this acronym. We are happy to use FLEX.

5) I don't see the QMC data points from Rost 2012 in Fig. 4. Were they forgotten?

It is present but lies on top of the point for Vekic 1993. That makes it hard to tell them apart. We will try if using different symbols helps.

---

## Round 1 · Referee Report · Anonymous (Referee 2) · 2020-7-14

Strengths

The manuscript addresses properties of the 2D Hubbard model, e.g. phase diagram, self-energy and spectral functions, which are still actively discussed. The authors propose new fRG approach, which yield reasonable results despite considering static vertices and one-loop approximation.

Weaknesses

Some parts of the text (Introduction, discussion of the results) need to be extended (see Report).

Report

The paper analyses phase diagram of 2D Hubbard model. The method and the results are certainly interesting, but some points require clarification.

Requested changes

  • The Introduction looks a bit too technical to me. It would be more preferable if the authors overview previous results on 2D Hubbard model, describe open problems, possible applications, etc.

  • Is there physical justification for the determination of Z-factors from the energy region |w|>2T as shown in Fig. 2? What is accounted and what is neglected by this approximation ? (see also next two points).

  • In Fig. 4 it would be interesting to see the comparison of the obtained critical scale to the results of DDMC approach of Phys. Rev. Lett. 124, 017003 (2020), together with the discussion of the nodal/antinodal dichotomy (see also the next point).

  • In Fig. 5 the authors show spectral functions, which are split (pseudoagp-like) in the nodal direction, but more quasiparticle like in the antinodal direction. Naively, I would expect the opposite (see DDMC results). The authors assign splitting in the nodal direction to thermal effects, which are similar to SOPT. However, since the considered (U,T) point is close to the crossover line, it is not clear why the nodal spectral function is not affected by correlations. Does that mean that the considered approach underestimates the effect of correlations on the spectral functions in the nodal direction (especially in view of agreement of the obtained crossover line to other approaches, including DDMC, where nodal and antinodal pseudogap appear at close values of T, U)?

  • In Fig. 6 the authors plot Z factors as a function of the angle around the Fermi surface. Is the Z factor determined in the way shown in Fig. 2? How the true quasiparticle residue (or, better, dReSigma/d\omega if it is positive) determined from small energy range |w|<<T, looks as a function of the angle around the Fermi surface?

  • What are the U, T parameters in Fig. 7? Where does the estimate U\approx 1.6 on the top of p. 9 come from?

  • validity: good
  • significance: high
  • originality: high
  • clarity: ok
  • formatting: good
  • grammar: excellent

Author:  Daniel Rohe  on 2020-10-20  [id 1013]

(in reply to Report 2 on 2020-07-14)
Category:
remark
correction

We thank the referee for these valuable and critical comments. Please find our replies to the individual points below.

  • The Introduction looks a bit too technical to me. It would be more preferable if the authors overview previous results on 2D Hubbard model, describe open problems, possible applications, etc.

We have extended the introduction in this area. Yet, this work is intentionally on the technical side and a comprehensive overview over open problems and possible applications is beyond of what we can offer here.

  • Is there physical justification for the determination of Z-factors from the energy region |w|>2T as shown in Fig. 2? What is accounted and what is neglected by this approximation ? (see also next two points).

We added a further statement on this in the main text. The approach is pragmatic, not rigorous. The physical justification, as stated in the text, consists in the observation that the spectral functions - albeit stemming from a non-Fermi-liquid-like self-energy in the strict sense - can well be approximated by a Fermi-liquid-like parametrisation in a certain range of T and U.

  • In Fig. 4 it would be interesting to see the comparison of the obtained critical scale to the results of DDMC approach of Phys. Rev. Lett. 124, 017003 (2020), together with the discussion of the nodal/antinodal dichotomy (see also the next point).

We have included the fit at the anti-node from Phys. Rev. Lett. 124, 017003 (2020) in figure and citation in caption of Uc_T

(Issue nodal/anti-nodal see reply to next point)

  • In Fig. 5 the authors show spectral functions, which are split (pseudoagp-like) in the nodal direction, but more quasiparticle like in the antinodal direction. Naively, I would expect the opposite (see DDMC results). The authors assign splitting in the nodal direction to thermal effects, which are similar to SOPT. However, since the considered (U,T) point is close to the crossover line, it is not clear why the nodal spectral function is not affected by correlations. Does that mean that the considered approach underestimates the effect of correlations on the spectral functions in the nodal direction (especially in view of agreement of the obtained crossover line to other approaches, including DDMC, where nodal and antinodal pseudogap appear at close values of T, U)?

We repaired the symbols in Fig. 4, added a Figure for Im(Sigma) and also a discussion of the angular dependence.

  • In Fig. 6 the authors plot Z factors as a function of the angle around the Fermi surface. Is the Z factor determined in the way shown in Fig. 2?

Yes. We added this in the caption

  • How the true quasiparticle residue (or, better, dReSigma/d\omega if it is positive) determined from small energy range |w|<<T, looks as a function of the angle around the Fermi surface?

There is no "true quasi-particle residue" at omega=0 since there is no quasi-particle when the slope is positive. We could analyse the slope of Re(Sigma) in this region, but we feel it does not add information beyond what is seen in Figure 3 and Figure 5.

  • What are the U, T parameters in Fig. 7? Where does the estimate U\approx 1.6 on the top of p. 9 come from?

The x-label ist wrong: The x-axis shows U, not \omega. Temperature is T=0.1 throughout the whole paper, unless indicated otherwise. In turn, the estimate U \approx 1.6 comes from the fact that on the x-axis (which should display U) the flow of the derivative of the self energy along the Fermi surface starts to deviate. That is when numerical criteria for the quality of the approximation begin to worsen, as outlined in the appendix.

We have corrected the x-axis label and added T=0.1 in the caption.

---

## Round 1 · Referee Report · Anonymous (Referee 3) · 2020-7-31

Report

This article presents a method that extends the standard one-loop functional renormalization group approach by including quasiparticle-based feedback on the flow and allows the direct determination of the real frequency dependence of the single-particle self-energy.

Results are presented for the frequency dependence of the single-particle spectral function compared with second order perturbation theory, as well as the crossover scale between weak and strong correlations, compared with various other implementations of the fRG.

This work is one of the few examples which presents real frequency behavior obtained with fRG and thus will be of interest to the community. I believe that this work potentially opens a new pathway for future work in the study of correlated electron systems and therefore recommend publication, but ask the authors to address the following points:

(1) The determination of the quasiparticle weight Z factors by neglecting the change in the low frequency slope of the real part of the self-energy is concerning. Can the authors come up with a better physical justification for this approximation?

(2) The spectral function in Fig. 5 seems to indicate the opening of a pseudogap at the node, and a quasiparticle peak at the anti-node. This is contrary to the results from other numerical studies of the half-filled Hubbard model. For recent work co-authored by one of the present authors, see e.g. arXiv:2006.10769. There it is shown, within numerically exact techniques, that a pseudogap first opens at the anti-node, and then at the node.

(3) Related to (2), arXiv:2006.10769 also presents fRG results for the self-energy on the imaginary frequency axis. Down to the temperatures accessible by fRG, these results are shown to compare well with the numerically exact QMC results. To better assess the quality of the present fRG scheme, I think it would be useful to add a figure showing results for the imaginary frequency behavior of the self-energy. This should be relatively easy to obtain from the current real frequency results and could potentially resolve the issue in (2).
  • validity: -
  • significance: -
  • originality: -
  • clarity: -
  • formatting: -
  • grammar: -

Author:  Daniel Rohe  on 2020-10-20  [id 1014]

(in reply to Report 3 on 2020-07-31)

We thank the referee for these valuable and critical comments. Please find our replies to the individual points below. Concerning the comparison to other methods, we point to arXiv:2006.10769, as remarked by the referee, since the essential requests made here are presented in that work.

  1. The determination of the quasiparticle weight Z factors by neglecting the change in the low frequency slope of the real part of the self-energy is concerning. Can the authors come up with a better physical justification for this approximation?

(Cf. reply to second report): We added a further statement on this in the main text. The approach is pragmatic, not rigorous. The physical justification, as stated in the text, consists in the observation that the spectral functions - albeit stemming from a non-Fermi-liquid-like self-energy in the strict sense - can well be approximated by a Fermi-liquid-like parametrisation in a certain range of T and U.

  1. The spectral function in Fig. 5 seems to indicate the opening of a pseudogap at the node, and a quasiparticle peak at the anti-node. This is contrary to the results from other numerical studies of the half-filled Hubbard model. For recent work co-authored by one of the present authors, see e.g. arXiv:2006.10769. There it is shown, within numerically exact techniques, that a pseudogap first opens at the anti-node, and then at the node.

As stated in the text, we do not observe any kind of pseudogap in the resulting quantities, neither at the node nor at the anti-node. The thermal dip is unrelated to pseudogap features induced by strong correlations, as in the sense of arXiv:2006.10769. Concerning the angular dependence of the thermal dip, we added a figure for Im(Sigma) as well as a discussion on the angular dependence of correlation effects and the thermal dip in the spectral function.

  1. Related to (2), arXiv:2006.10769 also presents fRG results for the self-energy on the imaginary frequency axis. Down to the temperatures accessible by fRG, these results are shown to compare well with the numerically exact QMC results. To better assess the quality of the present fRG scheme, I think it would be useful to add a figure showing results for the imaginary frequency behavior of the self-energy. This should be relatively easy to obtain from the current real frequency results and could potentially resolve the issue in (2).

See also reply to remark 2. SOPT results and fRG results from this method do not differ substantially on the real axis. Thus, they will not even more so on the imaginary axis. Data for SOPT on the imaginary axis are presented with higher accuracy and resolution in arXiv:2006.10769.

---

## Round 2 · Referee Report · Anonymous (Referee 3) · 2020-11-6

Strengths

This article presents a method that

  1. extends the standard one-loop functional renormalization group approach by including quasiparticle-based feedback on the flow, and

  2. allows the direct determination of the real frequency dependence of the single-particle self-energy.

  3. This work is one of the few examples which presents real frequency behavior obtained with fRG and thus will be of interest to the community.

Report

The authors' response to my previous comments is satisfactory. Publication can now be recommended.

---

## Round 2 · Referee Report · Anonymous (Referee 2) · 2020-11-12

Report

The authors have corrected most part of the points which raised the critique of the Referees. I would suggest only technical corrections as mentioned below.

Requested changes

I suggest to replot Fig. 4 where the plot takes only 1/4 of the space of the figure. E.g., the legend can be made two column and/or put to the caption to give more space to the plot.

In Fig. 7 the vertical axis can be restricted by the interval 0.85-0.95 (or similar) to give more space for the plot.

The horizontal axis in Fig. 8 can be restricted by 2.25 giving also more space for the plot.

  • validity: good
  • significance: high
  • originality: high
  • clarity: good
  • formatting: good
  • grammar: excellent

Author:  Daniel Rohe  on 2020-11-20  [id 1051]

(in reply to Report 2 on 2020-11-12)

We thank the referee for these helpful final suggestions which will be implemented.

---

## Round 2 · Referee Report · Walter Metzner (Referee 1) · 2020-11-18

Report

The authors have replied satisfactorily to my questions and comments. The provided references on other real frequency fRG calculations are enough. Others can be found in the review by Dupuis et al. which they also cite.

I recommend publication in the present form.

---

## Round 2 · List of Changes

• added content in reply to referee one -> see reply to referee
  • added content in reply to referee two -> see reply to referee
  • added content in reply to referee three -> see reply to referee

---

## Editorial Decision

published